# Predicting a Protein's Stability
# under a Million Mutations

**Jeffrey Ouyang-Zhang**
UT Austin
jozhang@utexas.edu

**Daniel J. Diaz**
UT Austin
danny.diaz@utexas.edu

**Adam R. Klivans**
UT Austin
klivans@cs.utexas.edu

**Philipp Krähenbühl**
UT Austin
philkr@cs.utexas.edu

## Abstract

Stabilizing proteins is a foundational step in protein engineering. However, the evolutionary pressure of all extant proteins makes identifying the scarce number of mutations that will improve thermodynamic stability challenging. Deep learning has recently emerged as a powerful tool for identifying promising mutations. Existing approaches, however, are computationally expensive, as the number of model inferences scales with the number of mutations queried. Our main contribution is a simple, parallel decoding algorithm. Our *Mutate Everything* is capable of predicting the effect of all single and double mutations in one forward pass. It is even versatile enough to predict higher-order mutations with minimal computational overhead. We build our *Mutate Everything* on top of ESM2 and AlphaFold, neither of which were trained to predict thermodynamic stability. We trained on the Mega-Scale cDNA proteolysis dataset and achieved state-of-the-art performance on single and higher-order mutations on S669, ProTherm, and ProteinGym datasets. Our code is available at https://github.com/jozhang97/MutateEverything.

## 1 Introduction

Protein engineering is the discipline of mutating a natural protein sequence to improve properties for industrial [5, 78] and pharmaceutical applications [2, 25, 41]. However, evolution simultaneously optimizes several properties of a protein within its native environment, resulting in proteins with marginal thermodynamic stability ($\sim$ 5-15 kcal/mol) [39] which become non-functional in an industrial setting. Therefore, repurposing a natural protein for biotechnological applications usually begins with identifying non-deleterious mutations that stabilize the structure. With a stabilized structure, downstream engineering goals, which often require exploring destabilizing mutations, become tractable. Historically, this process has been bottlenecked by the requirements of extensive laboratory characterization of rational designs [33, 79] or directed evolution libraries [4, 12, 21, 22]. The recent explosion in biological data [9, 29, 72, 73] has enabled deep learning frameworks to accelerate the identification of stabilizing mutations. A successfully stabilized protein often requires several mutations. However, current frameworks do not take into account epistatic interactions between multiple mutations. Thus, to significantly accelerate protein engineering, it is critical to efficiently navigate the epistatic landscape of higher-order mutations [10, 69]. However, due to its combinatorial nature, thorough exploration quickly becomes computationally prohibitive.

In this paper, we introduce *Mutate Everything* to directly predict changes in thermodynamic stability ($\Delta\Delta G$) for all single and higher-order mutations jointly. *Mutate Everything* is a parallel decoding algorithm that works in conjunction with a sequence model for predicting thermodynamic stability.

37th Conference on Neural Information Processing Systems (NeurIPS 2023).

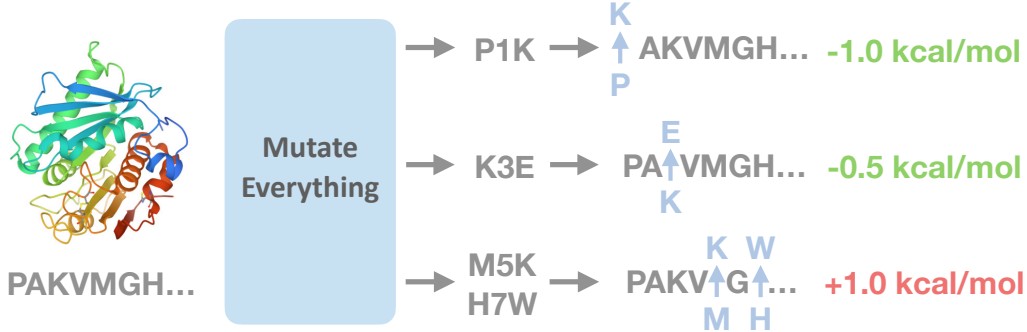

Figure 1: *Mutate Everything* efficiently predicts $\Delta\Delta G$, the change in thermodynamic stability of folding, for over a million mutations (e.g. all single, double mutations) in a single inference step. This helps identify and prioritize stabilizing mutations ($\Delta\Delta G < 0$) in protein engineering efforts. The notation for a mutation is $AA_{from}$, pos, $AA_{to}$ (e.g. "P1K" mutates from P to K at position 1)

Prior models for predicting thermodynamic stability work by producing an embedding or representation of the input sequence. Our decoder takes this representation and uses a linear network to create further representations, one for every possible single mutation. These mutation-level representations are then further aggregated to form representations for higher-order mutations. We feed these higher-order representations into a lightweight multi-layer perception (MLP) head to output predicted $\Delta\Delta G$ measurements. Since the mutation-level representations are computed only once, we are able to scale to millions of (higher-order) mutations, as the aggregation and MLP-head computations are inexpensive. As such, given a fixed computational budget, our model evaluates millions more potential mutations than prior methods.

*Mutate Everything* estimates the effects of all single and double mutations for a single protein in seconds on one GPU. This can efficiently compute the stability of *all* single and double mutants across *all* ∼20,000 proteins in the human proteome within a short time frame. To the best of our knowledge, *Mutate Everything* is the first tool that renders the computational analysis of double mutants across the entire proteome tractable. *Mutate Everything* can be used in conjunction with any model that generates representations for protein sequences. In this paper, we use AlphaFold [31] as a backbone architecture and fine-tune it on a dataset labeled by experimentally derived $\Delta\Delta G$ measurements. This results in the first accurate model of stability prediction based on AlphaFold.

We evaluate the performance of our *Mutate Everything* on well-established benchmarks for predicting experimentally-validated $\Delta\Delta G$ measurements of mutations: ProTherm [48], S669 [54], and ProteinGym [49]. On ProTherm high-order mutations (PTMul), our model achieves a Spearman of 0.53, compared to 0.50 in the next best method. On S669 single mutations, our model achieves a Spearman of 0.56, outperforming the state-of-the-art at 0.53. On ProteinGym, the *Mutate Everything* outperforms state-of-the-art methods from 0.48 to 0.49. Our model makes finding the most stabilizing double mutations computationally tractable. We show on the double mutant subset of the cDNA-proteolysis dataset (cDNA2) [72], where only 3% of known mutations are stabilizing, that *Mutate Everything* effectively ranks stabilizing mutations ahead of destabilizing ones.

## 2   Related Works

### 2.1   Protein Engineering

Protein engineering is the process of mutating a natural protein to improve particular phenotypes, such as thermodynamic stability and function. The field has historically relied on rational design and stochastic methods, such as error-prone PCR [1], DNA shuffling [71], and directed evolution (DE) [4, 27], to identify gain-of-function mutations. Rational design is limited to proteins with solved structures and requires an extensive understanding of biochemistry and specific knowledge of the particular protein to select mutations highly likely to improve the target phenotypes [33, 79]. Directed evolution requires little to no knowledge of the protein and instead generates a library of protein variants that are then screened for the target phenotypes (e.g. fluorescence brightness, antibiotic

resistance, stability, activity) [4, 27]. The library can be generated via site-saturated mutagenesis of a handful of positions in the sequence [12] or DNA shuffling, in which the gene is digested into random fragments and reassembled into full-length sequences [71]. After screening the library, the most "fit" variant is then selected as the initial sequence for the next round of directed evolution. This iterative process repeats until it obtains a protein variant with the desired phenotypes.

Machine learning has demonstrated its ability to augment rational design and accelerate the stabilization of a variety of proteins [20, 28, 40, 52, 68]. Separately, machine learning-guided directed evolution (MLDE)[75] has been shown to improve the likelihood of obtaining the global fitness maximum by 81-fold compared to traditional DE [76]. MLDE has accelerated the engineering of several proteins, such as the enantioselectivity of enzymes for kinetic resolution of epoxides [23] and the activity and expression of a glutathione transferase [46]. *Mutate Everything* empowers the experimentalist to accelerate the stabilization of a protein for both rational design and MLDE.

## 2.2 Machine Learning for Protein Structure Prediction

Recent advances in machine learning have led to remarkable progress in protein structure prediction. AlphaFold [31] has demonstrated that deep learning is highly effective at predicting protein structures from a sequence by using evolutionary history via a multiple sequence alignment (MSA). AlphaFold passes the MSA and a pairwise representation of the sequence into Evoformer to capture the co-evolutionary patterns between residues. The Evoformer output is then processed by the Structure Module, which predicts the protein's structure. We challenge prior works postulating that AlphaFold cannot be used for stability prediction and show that fine-tuning these rich co-evolutionary and structural features yield highly performant stability predictors [53].

Evolutionary Scale Modeling (ESM) [38, 62] has shown that protein structure prediction can be performed without MSAs and specialized architectures by leveraging large transformers pre-trained on masked token prediction. Other works extended this masked pre-training framework, including MSA-Transformer [60] which incorporates a sequence's MSA as input, and Tranception [49], which develops a hybrid convolution and attention based autoregressive architecture. We show that fine-tuning these evolutionary representations excels at protein stability assessment without MSAs.

## 2.3 Protein Stability Assessment

Traditional approaches to protein stability assessment relied on a combination of physics-based methods, statistical analysis, and traditional machine learning techniques. Physics-based tools, such as FoldX, Rosetta, and SDM, utilize energy functions and statistical patterns to assess how mutations affect a protein's stability [32, 67, 77]. DDGun [44] directly inferred $\Delta\Delta G$ from heuristics, including Blosum substitution scores, differences in interaction energy with neighboring residues, and changes in hydrophobicity. Many traditional machine learning approaches used support vector machines and decision trees with physics-based feature engineering [11, 14, 15, 17, 37, 66]. Others ensemble existing machine learning methods [35, 56, 59, 63, 64].

Recently, deep learning-based approaches have recently begun to outperform existing physics and traditional machine learning approaches [13, 19, 34]. ACDC-NN [6] trains an asymmetric convolutional neural network for predicting forward and reverse mutation effects. Thermonet [36] voxelizes and feeds both the wild-type and mutant protein structures into a 3D-CNN to regress a $\Delta\Delta G$ value. PROSTATA feeds the wild-type and mutant protein sequence into a pre-trained ESM2 model and then regresses a $\Delta\Delta G$ value [74]. Stability Oracle takes the native and mutant amino acid type along with the local structure as input and regresses $\Delta\Delta G$ [18]. Several deep learning frameworks [64, 80, 81] model multiple mutations in addition to single mutations. In this paper, we develop a framework that models the protein to enable efficient enumeration of all mutation candidates.

# 3 Preliminary

**Problem Setup.** A protein $w = (w_1, ..., w_L)$ is a sequence of amino acids $w_l \in AA$, where $AA = \{A, C, ..., Y\}$ are the 20 different amino acid types encoded by genetic information. Let $\mu = (p, t)$ denote a mutation substituting the amino acid $w_p$ at position $p \in [1, ..., L]$ to amino acid type $t \in AA$. Our goal is to determine the change in thermodynamic stability $\Delta\Delta G \in \mathbb{R}$ for a protein $w$ under a large number of mutation sets $(M_1, ..., M_N)$ where a mutation set $M_i = \{\mu_k\}_{k=1}^{K_i}$.

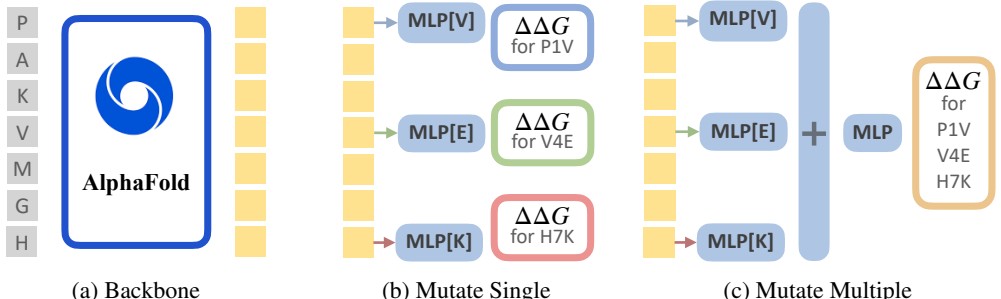

| (a) Backbone | (b) Mutate Single | (c) Mutate Multiple |

Figure 2: **Mutate Everything decodes any mutation.** The model takes a protein sequence of length $L$ as input. (a) A pre-trained feature extractor (e.g. ESM [38], AlphaFold [31]) computes per-position features for each amino acid in the sequence (yellow). Each token is reshaped by an MLP into an array of all possible mutations at that location. (b) A single mutation is directly decoded to its $\Delta\Delta G$. (c) For a mutation set containing multiple mutations, selected mutations are aggregated and decoded to obtain a predicted $\Delta\Delta G$.

The mutation sets include higher-order mutations ($K > 1$), which is beyond the scope of state-of-the-art single mutation predictors ($K = 1$) [18, 36, 74]. These higher-order mutations are strictly more challenging than single mutations. The model must understand the effect of each single mutation and also the epistatic interactions that arise between individual mutations. We introduce a method that captures these interactions for any mutation set and decodes a corresponding $\Delta\Delta G$.

Furthermore, existing paradigms cannot efficiently scale to a large number of mutations. Higher-order mutation models [16, 35, 44, 64] take a protein and its mutant as input and predict the change in stability. These methods would need to enumerate and sort over a million model evaluations to find the most stabilizing mutations for a small protein of 75 amino acids. Our model runs a fine-tuned backbone only once and a lightweight decoder once for each mutation in parallel.

**Feature Extraction.** We fine-tune AlphaFold and other pre-trained models to alleviate the scarcity of labeled data. AlphaFold's Evoformer learns co-evolutionary patterns, such as residue interactions, and the Structure Module learns structural information, such as solvent accessibility or residue-residue distance. We demonstrate this is connected to a protein's stability and how mutations affect stability.

Each element of the protein sequence is tokenized by its amino acid type $w_l$ and positionally encoded. This produces a set of high-dimensional tokens for the transformer to take as input. The tokens are fed into a pre-trained model, which extracts per-position features $(x_l)_{l=1}^{L}$ where $x_l \in \mathbb{R}^d$. The transformer feature extractor is finetuned with our training objective. This is illustrated in Figure 2a.

# 4 Methodology

Our model uses per-position features $(x_l)_{l=1}^{L}$ to determine the stability of a mutation set $M$. We distinguish between single point mutations $K = |M| = 1$, and higher-order mutations $K > 1$.

Both single and higher-order mutations share the same backbone architecture and encoding scheme, see Figure 2b. For each mutation $\mu = (p, t)$ at position $p$ to amino acid $t$, we compute a latent mutation representation $z(\mu) \in \mathbb{R}^d$, which captures the effect of the mutation on the protein. We decompose the mutation representation $z(\mu) = f^t(x_p) + h^t$, where sequence-dependent features $f^t \in \{f^A, ..., f^Y\}$ project the per-position feature according to the mutated amino acid type, and sequence-independent features $h^t \in \{h^A, ...h^Y\}$ are embeddings unique to the amino acid types $A, C, ..., Y$. Intuitively, $f^t$ captures a contextualized feature at the position, and $h^t$ captures information about the mutated amino acid type. This mutation representation builds up a feature representing the amino acid substitution.

**Decoding single mutations.** Figure 2b illustrates single mutation decoding. The change in thermodynamic stability $\Delta\Delta G$ for a single mutation $\mu$ is read out from its mutation representation $z(\mu)$. A lightweight linear head $g^1 : \mathbb{R}^d \rightarrow \mathbb{R}$ decodes this representation into the change in thermodynamic stability: $\Delta\Delta G^{pr} = g(z(\mu))$.

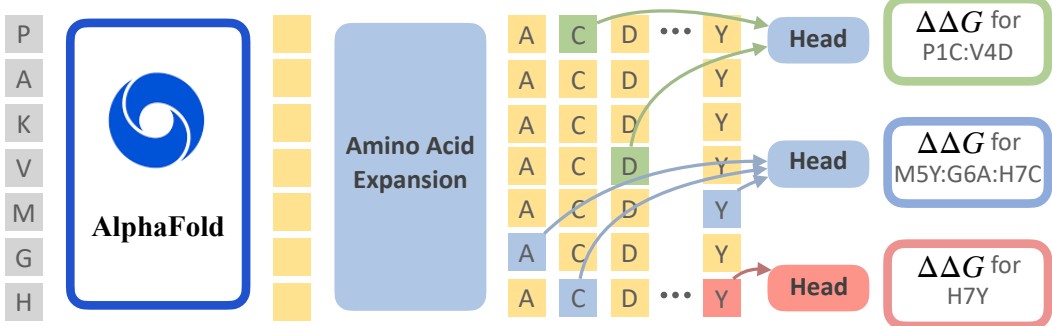

Figure 3: **Mutate Everything decodes many mutations.** Each token output by the backbone $x_l$ is expanded to 20 tokens $z(\mu)$, each corresponding to one unique amino acid type (yellow with amino acid types). For each mutation set $M$, we first aggregate the features corresponding to each mutation's position and amino acid being mutated to (arrows). A lightweight head $g$ then decodes the aggregated features to obtain the predicted $\Delta\Delta G$. These heads are illustrated in Figure 2b and Figure 2c. Parallel feature aggregation and decoding enable efficient stability prediction for millions of mutations.

**Decoding higher-order mutations.** Figure 2c illustrates higher-order mutation decoding. The $\Delta\Delta G$ for a higher-order mutation $M = \{\mu_k\}_{k=1}^{K}$ is read out from its $K$ corresponding single point mutation representations $\{z(\mu_k)\}_{k=1}^{K}$. *Mutate Everything* aggregates the representations for each single point mutation within a higher-order mutation. By aggregating the single mutations in the latent space, the model learns to account for the interactions between them. We aggregate with summation for its simplicity. A lightweight head $g : \mathbb{R}^d \rightarrow \mathbb{R}$ then decodes the higher-order mutation $\Delta\Delta G^{pr} = g(\sum_k z(\mu_k))$. This formulation allows us to seamlessly parameterize any higher-order mutation. For higher-order mutations (e.g. $K > 1$), we empirically find that it is beneficial to predict the residual to the sum of single mutation $\Delta\Delta G$s from our single mutation decoder (see Table 5d).

**Decoding a million mutations in one forward pass.** This decoding scheme efficiently compute $\Delta\Delta G$ for *millions* of mutations, whether single or higher-order (see Figure 3). Our insight is to precompute all mutation features and reuse them for all predictions. First, we compute all $L \times 20$ mutation representations $Z = \{z((p, t)) : p \in [1, ..., L], t \in AA\}$. This expands each token $x_l$ to 20 representations that each capture the effect of mutating to one specific amino acid (including the wild-type amino acid). The backbone and this expansion are executed once for a protein sequence.

A lightweight head decodes $\Delta\Delta G$ for any higher-order mutation (a mutation set). For each mutation set, we index and add the relevant mutation representations in $Z$. Index and sum operations are parallelized with outer-summation for $K = 2$ and the "gather" operation for $K > 2$. Finally, the lightweight decoder $g$ computes the $\Delta\Delta G$ for each mutation set in parallel.

**Training.** Our model is fine-tuned using supervised learning. We optimize the Huber loss.

$$\mathcal{L}(w, (M_i, \Delta\Delta G_i^{exp})_{i=1}^{N}) = \frac{1}{N} \sum_{i=1}^{N} ||\Delta\Delta G^{pr}(w, M_i) - \Delta\Delta G_i^{exp}||_H \qquad (1)$$

where $w$ is a protein sequence and $(M_i, \Delta\Delta G_i^{exp})_{i=1}^{N}$ are the mutation sets for this protein with experimentally validated $\Delta\Delta G$. This approach efficiently backpropagates the loss through *all* labeled mutations for a given protein in a single pass, unlike prior work which backpropagates through only one mutation in a single pass. We train on single and double mutants with on average 2000 mutations for each protein. When training on double mutants, we learn the residual to the sum of the experimental (ground truth) $\Delta\Delta G$ for the constituent single mutations. Note that we use the *experimental* $\Delta\Delta G$ of the single mutants to train double mutants but during testing, we use the model's *predicted* $\Delta\Delta G$ of the single mutants to infer double mutant $\Delta\Delta G$. Thus, we do not need experimental data for single effects to predict combined effects.

Only 3% of the double mutants in our training set are stabilizing ($\Delta\Delta G < $ -0.5 kcal/mol [8]). To improve our performance on stabilizing mutations, we subsample the number of destabilizing double mutations to at most 8 times the number of stabilizing double mutations.

| Method | Stabilizing $r_s$ | nDCG | DetPr | AUC | MCC |
|---|---|---|---|---|---|
| Mean | 0.00 | 0.02 | 0.03 | 0.50 | 0.00 |
| MSA | 0.04 | 0.06 | 0.02 | 0.60 | 0.01 |
| ESM2 [38] | 0.04 | 0.08 | 0.02 | 0.65 | 0.00 |
| PROSTATA [74] (Additive) | 0.08 | 0.15 | 0.05 | 0.75 | 0.05 |
| *Mutate Everything* (Additive) | 0.08(0.03) | 0.25(0.01) | 0.10(0.01) | 0.81(0.01) | 0.25(0.02) |
| *Mutate Everything* (Ours) | **0.14**(0.02) | **0.43** (0.02) | **0.16** (0.01) | **0.84** (0.01) | **0.27** (0.01) |

Table 1: **Stabilizing Double Mutation Results on cDNA2.** The metrics evaluate the model's performance on stabilizing mutations. Our additive baseline naively adds predicted single mutation $\Delta\Delta G$s. Our model is state-of-the-art at finding stabilizing mutations. The proteins in cDNA2 val have at most 36% sequence similarity to those in the training set. Parenthesis indicates standard error across 7 training runs.

## 5 Results

### 5.1 Implementation Details

Our primary feature extractor is AlphaFold, implemented in OpenFold [3, 31]. The multiple sequence alignment (MSA) is computed using Colabfold [43]. Sequence features from the Evoformer and Structure Module are aggregated as input to the decoder. An adapter maps the backbone hidden dimension to $D = 128$. Our amino acid projections and single mutation decoders are implemented with a linear layer. The higher-order mutation decoder transforms the previous embedding with a 2-layer MLP. These representations are aggregated and fed into a 3-layer MLP to predict $\Delta\Delta G$.

We train on the **cDNA** display proteolysis dataset [72], which leverages a high throughput proteolysis screen enabled by next generation sequencing (NGS) to extract noisy $\Delta\Delta G$ values for 100 mini-proteins totaling over 100,000 single and double mutations. To evaluate generalization on unseen proteins, we remove proteins from our training set that have high sequence similarity using MMSeqs2 [70] to any protein in our evaluation benchmarks. The short protein lengths ($< 100$ amino acids) keep the memory requirement small for the higher-order decoder. See Section A.1 for more information about **cDNA** and other datasets.

We fine-tune a pre-trained backbone on single mutations for 20 epochs. Then, we finetune the model on both single and double mutations for 100 epochs using a cosine learning rate schedule with 10 warmup epochs. We use a batch size of 3 proteins due to the high memory requirements of AlphaFold. We use a learning rate of $3 \times 10^{-4}$ and weight decay of 0.5. Training takes 6 hours on 3 A100 GPUs.

### 5.2 Finding Stabilizing Double Mutations

We evaluate our model's ability to find the stabilizing mutations on cDNA2. **cDNA2** is our validation split of the cDNA double mutations dataset, consisting of 18 mini-proteins totaling 22,000 double mutations. The proteins in the validation set have at most 36% homology with those in the cDNA training set. Of these mutations, only 198 or 0.8% are stabilizing with $\Delta\Delta G < -0.5$ kcal/mol [8]. Like protein engineering, this scenario closely resembles the challenge of identifying a small number of stabilizing mutations amidst a much larger pool of destabilizing or neutral mutations.

We evaluate different methods on *stabilizing* mutations in Table 1. Stabilizing $r_s$ is the Spearman coefficient on the experimentally stabilizing subset introduced in [65]. Normalized discounted cumulative gain (nDCG) [30] measures the quality of the ranked mutation list by taking into account its experimentally validated $\Delta\Delta G$ and position in the list [59]. Detection Precision (DetPr) is the proportion of experimentally validated stabilizing mutations among the top $K = 30$ predictions. We adapt previous state-of-the-art single mutation predictors by naively adding both single mutation predictions. A detailed description of the metrics is found in Section A.2.

Our model demonstrates exceptional performance in prioritizing stabilizing double mutations over destabilizing ones, achieving a significantly higher normalized discounted cumulative gain of 0.43 compared to 0.25, as well as a superior detection precision of 0.16 compared to 0.10. Our model additionally improves classification metrics Matthews Correlation Coefficient (MCC) and Area under Precision-Recall Curve (AUC) by 0.02 and 0.03, respectively.

| Method | $r_s$ | AUC | MCC | RMSE $_\downarrow$ | Stabilizing $r_s$ |
|---|---|---|---|---|---|
| Mean (cDNA) | 0.00 | 0.50 | 0.00 | 2.42 | 0.00 |
| Mean (ProTherm) | 0.00 | 0.50 | 0.00 | 2.26 | 0.00 |
| MSA | 0.07 | 0.51 | -0.05 | N/A | 0.02 |
| ESM2 [62] | 0.05 | 0.51 | 0.04 | N/A | -0.01 |
| FoldX [67] | 0.41 | - | - | 2.95 | - |
| DDGun [44] | 0.25 | 0.63 | 0.16 | 2.21 | 0.17 |
| DDGun3D [44] | 0.26 | 0.64 | 0.18 | 2.24 | 0.17 |
| PROSTATA [74] (Additive) | 0.21 | 0.60 | 0.05 | 2.25 | 0.00 |
| *Mutate Everything* (Additive) | 0.50(0.02) | 0.76(0.01) | 0.37(0.02) | **2.02**(0.03) | 0.20(0.01) |
| *Mutate Everything* (Ours) | **0.53**(0.01) | **0.78**(0.01) | **0.43**(0.01) | 2.04(0.01) | **0.19**(0.01) |

Table 2: **Multiple Mutation Results on PTMul.** Our additive baseline naively adds predicted single mutation $\Delta\Delta G$s. Our model presents a strong mutation assessor. The proteins in PTMul have at most 35% sequence similarity to those in the training set. Parenthesis is standard error across 7 runs.

To the best of our knowledge, *Mutate Everything* is the first work that models all double mutations in a computationally tractable manner. Figure 4 shows the runtime of several methods on a protein of 317 amino acids on an A100 GPU. The dashed line indicates the transition from evaluating single mutants to double mutants. *Mutate Everything* predicts $\Delta\Delta G$ for all single and double mutations in one pass of the model. It runs in 0.6 seconds using an ESM2 backbone, and 12.1 seconds on an AlphaFold backbone. PROSTATA [74] also uses the ESM2 backbone but takes a protein sequence and its mutated sequence as input and outputs the change in stability. PROSTATA takes 306 hours to evaluate all double mutations with a batch size of 768. On an 8 GPU node, this will take 1.5 days.

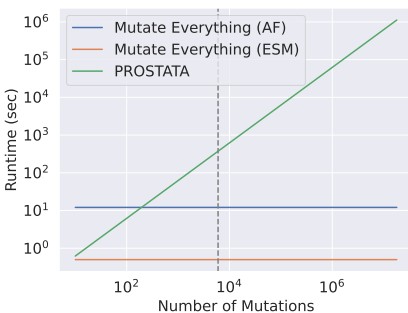

Figure 4: **Runtime analysis**. Our model's runtime is constant.

## 5.3 Higher-order Mutation Results

We evaluate our model for predicting changes in thermodynamic stability for higher-order mutations in ProTherm. ProTherm is a database of thermodynamic ($\Delta\Delta G$) and thermostability ($\Delta T_m$) experimental characterization of protein mutations curated from the literature. We consider a subset of this database that contains $\Delta\Delta G$ values for higher-order mutations, named ProTherm Multiple (**PTMul**). PTMul contains 858 mutations [44]. We keep 846 experimentally validated $\Delta\Delta G$ for higher-order mutants, including 536 doubles and 183 triples, after removing 4 duplicate and 8 ambiguous mutations. PTMul proteins have at most 35% homology to the proteins used in training.

Table 2 shows our results on PTMul. A description of baselines is provided in Section A.3. Our additive baselines naively add the sum of single mutation $\Delta\Delta G$s. This baseline rigorously evaluates our model's ability to learn epistasis, or the non-additive interactions when combining two mutations. Mean always predicts the global test dataset mean statistic. ESM and MSA perform poorly when comparing mutations across proteins. On ProTherm, *Mutate Everything* achieves the state-of-the-art performance of 0.53 Spearman correlation $r_s$, compared to 0.50 of our additive baseline. A breakdown of PTMul performance on double mutations and more ($> 2$) mutations is found in Table 6.

While other methods also handle multiple mutations, they usually adopt a mutation- or protein-level train and test split. This results in data leakage since the same protein or a homolog will have mutations in both the training and testing sets, leading to inflated reported metrics [16, 18, 35, 64]. In our study, we use sequence similarity-based splitting where our training proteins have at most 35% homology to those in PTMul. To fairly evaluate generalization to new proteins, we exclude these inflated comparisons from our study.

| Method | Forward | | | | Reverse | | | |
| --- | --- | --- | --- | --- | --- | --- | --- | --- |
| | $r_s$ | AUC | MCC | RMSE$_\downarrow$ | $r_s$ | AUC | MCC | RMSE$_\downarrow$ |
| **Structure-based Methods** | | | | | | | | |
| mCSM [57] | 0.37 | 0.66 | 0.13 | 1.53 | 0.24 | 0.61 | 0.10 | 2.33 |
| I-Mutant3.0 [11] | 0.35 | 0.64 | 0.08 | 1.53 | 0.17 | 0.59 | 0.06 | 2.35 |
| DUET [56] | 0.42 | 0.68 | 0.19 | 1.52 | 0.26 | 0.63 | 0.12 | 2.16 |
| FoldX [67] | 0.27 | 0.62 | 0.14 | 2.35 | 0.32 | 0.67 | 0.20 | 2.54 |
| MAESTRO [35] | 0.46 | 0.69 | 0.26 | 1.45 | 0.22 | 0.61 | 0.11 | 2.12 |
| PopMusic [17] | 0.42 | 0.69 | 0.22 | 1.51 | 0.24 | 0.62 | 0.12 | 2.10 |
| SDM [77] | 0.39 | 0.67 | 0.21 | 1.67 | 0.14 | 0.59 | 0.12 | 2.15 |
| INPS3D [66] | 0.44 | 0.70 | 0.20 | 1.49 | 0.36 | 0.69 | 0.23 | 1.78 |
| Dynamut [63] | 0.38 | 0.68 | 0.20 | 1.59 | 0.37 | 0.67 | 0.17 | 1.69 |
| ThermoNet [36] | 0.38 | 0.69 | 0.21 | 1.62 | 0.35 | 0.66 | 0.18 | 1.66 |
| PremPS [14] | 0.42 | 0.66 | 0.20 | 1.50 | 0.43 | 0.66 | 0.22 | 1.49 |
| DDGun3D [44] | 0.43 | 0.71 | 0.27 | 1.60 | 0.41 | 0.71 | 0.23 | 1.61 |
| ACDC-NN [7] | 0.46 | 0.73 | 0.23 | 1.49 | 0.45 | 0.72 | 0.22 | 1.50 |
| Stability Oracle [18] | **0.53** | **0.75** | **0.34** | **1.44** | **0.53** | **0.75** | **0.32** | **1.43** |
| **Sequence-based Methods** | | | | | | | | |
| MuPro [15] | 0.27 | 0.61 | 0.03 | 1.60 | 0.22 | 0.64 | 0.08 | 2.41 |
| I-Mutant3.0-Seq [11] | 0.34 | 0.67 | 0.25 | 1.54 | 0.23 | 0.61 | 0.05 | 2.25 |
| DDGun [44] | 0.43 | 0.72 | 0.24 | 1.73 | 0.41 | 0.71 | 0.23 | 1.76 |
| ACDC-NN-Seq [55] | 0.44 | 0.72 | 0.26 | 1.52 | 0.44 | 0.72 | 0.23 | 1.52 |
| INPS-Seq [66] | 0.44 | 0.72 | 0.25 | 1.52 | 0.44 | 0.72 | 0.21 | 1.53 |
| PROSTATA [74] | 0.50 | 0.73 | 0.28 | 1.44 | **0.50** | **0.73** | 0.29 | **1.44** |
| *Mutate Everything* (Ours) | **0.56** | **0.76** | **0.35** | **1.38** | 0.49 | **0.73** | **0.30** | 1.48 |

Table 3: **Comparisons on Single Mutation Thermodynamic Stability prediction in S669.** We report regression and classification metrics on the forward and reverse datasets for structure-based approaches (top block) and sequence-based approaches (bottom block). r refers to Spearman correlation coefficient. Our training proteins have at most 30% sequence similarity with those in S669. *Mutate Everything* is state-of-the-art on S669 forward.

## 5.4   Single Mutation Results

We additionally validate our model's ability to predict changes in thermodynamic stability under single mutations. We compare *Mutate Everything* to prior works on the newly introduced **S669** dataset, which was established to address the data leakage issue and enable fair comparisons between existing methods [54]. We also evaluate the commonly studied reverse dataset setting, in which the model takes the mutant sequence as input and predicts $\Delta\Delta G$ for mutating back to the wild-type sequence. The experimental value is obtained by negating the original $\Delta\Delta G$ value going from the wild-type sequence to the mutant sequence. S669 is well curated and extensive, containing 94 proteins totaling 669 mutations [54]. S669 proteins have at most 30% sequence similarity with those in our training set to ensure separation between the training and validation sets.

Table 3 shows our comparisons with prior works. *Mutate Everything* is state-of-the-art on S669, achieving a Spearman correlation of 0.56, where the prior art obtained 0.53. *Mutate Everything* outperforms the existing sequence model by a large margin, with a 6-point gain on Spearman correlation and a 3-point improvement on AUC. Our method is even as powerful as the latest structure-based methods which have access to atom coordinates.

*Mutate Everything* is competitive on the reverse dataset evaluation, in which the original and mutated sequences are flipped and $\Delta\Delta G$ negated. Our performance drops on reverse mutations because reverse mutations are out of distribution for our model. We did not perform data augmentation to train on reversed mutations as commonly done in the literature [18, 36, 74]. We found that it was beneficial to bias our reverse predictions using $\Delta\Delta G$ to the amino acid found in the original sequence. We found the performance to be similar across 5 runs ($< 0.01$ standard error).

| Model | ProteinGym-Stability | | | ProteinGym | | |
|---|---|---|---|---|---|---|
| | $r_s$ | AUC | MCC | $r_s$ | AUC | MCC |
| DeepSequence [61] | 0.44 | 0.74 | 0.34 | 0.43 | 0.74 | 0.35 |
| EVE [24] | 0.47 | 0.75 | 0.36 | 0.46 | 0.76 | 0.36 |
| ESM1v [42] | 0.41 | 0.72 | 0.31 | 0.41 | 0.74 | 0.33 |
| Progen2 [47] | 0.45 | 0.74 | 0.35 | 0.42 | 0.74 | 0.34 |
| MSA Transformer [60] | 0.50 | 0.77 | 0.38 | 0.44 | 0.75 | 0.35 |
| Tranception [49] | 0.40 | 0.71 | 0.31 | 0.40 | 0.73 | 0.32 |
| Tranception+MSA | 0.45 | 0.74 | 0.35 | 0.45 | 0.76 | 0.36 |
| TranceptEVE [50] | 0.50 | 0.77 | 0.38 | 0.48 | 0.77 | 0.38 |
| MSA | 0.41 | 0.72 | 0.32 | 0.39 | 0.72 | 0.31 |
| *Mutate Everything* | 0.52 | 0.78 | 0.39 | 0.38 | 0.72 | 0.30 |
| *Mutate Everything* +MSA | 0.52 | 0.78 | 0.41 | 0.44 | 0.75 | 0.36 |
| *Mutate Everything* +MSA+Tranception | **0.53** | **0.79** | **0.42** | **0.49** | **0.78** | **0.39** |

Table 4: **Comparisons on mutation fitness prediction in ProteinGym.** We report ranking and classification metrics on both a subset of ProteinGym with stability-like phenotypes and the entire ProteinGym dataset. Our stability model can generalize to all phenotypes, outperforming existing methods on stability-like phenotypes and performing similarly to existing methods otherwise. The maximum homology between our training set and ProteinGym proteins is 43%.

## 5.5 Generalization to other Phenotypes

We show that stability predictors generalize to other phenotypes as well. **ProteinGym** substitutions is a dataset of 87 proteins with 1.6 million mutation sets labeled with a generic fitness score, such as activity, growth, or expression. The mutation sets have up to 20 substitutions and 14 proteins contain higher-order mutations. The ProteinGym-Stability split contains 7 proteins and 26,000 mutations with fitness labels that are known to correlate with thermodynamic stability (thermostability, abundance, expression). ProteinGym proteins have at most 45% sequence similarity with those in our training set. Due to memory constraints, we truncate proteins to at most 2000 amino acids.

Table 4 compares our work with previous works. Our MSA baseline uses the empirical amino acid distribution in the MSA at a given position to compute a likelihood ratio. The MSA retrieval prior weights the scores using the per-position first-order statistics in a multiple sequence alignment. On ProteinGym-Stability, *Mutate Everything* obtains a Spearman correlation of 0.52, compared to 0.50 of the next best method, TranceptEVE. On the one protein with labeled stability, *Mutate Everything* obtains 0.51 vs 0.33 of Tranception with MSA retrieval [51]. On the full ProteinGym benchmark, we outperform the strongest methods by ensembling our predictions with Tranception [49]. Ensembling our method with evolutionary models by averaging scores gains 0.11 points on ProteinGym and 0.01 points on ProteinGym-Stability, suggesting that our model learns complementary information to the evolutionary models. For comparison, TransEVE ensembles an evolutionary model (Tranception) and a family-based model (EVE) for a 0.03 point gain [50]. We found the performance to be similar across 5 runs ($< 0.01$ standard error). An in-depth study on how protein stability related to function can be found in [26].

## 5.6 Ablation Studies

We ablate several architectural design choices in Table 5. Core architectural designs are evaluated on S669. Architectural designs for only multiple mutations are evaluated on cDNA2.

**Backbone**. We analyze the effect of different feature extractors on single mutation stability assessment in Table 5a. ESM2 [38] takes only sequence as input whereas MSA-Transformer [60] and AlphaFold [31] take the sequence and its MSA as input. Both ESM2 and MSA-Transformer are trained on masking residue modeling while AlphaFold is also trained on structure prediction. MSA input improves the stability prediction performance and AlphaFold's architecture and training further improve stability prediction.

| backbone | $r_s$ | AUC |
|---|---|---|
| ESM2 [38] | 0.47 | 0.72 |
| MSA-Transformer [60] | 0.53 | 0.75 |
| AlphaFold (scratch) | 0.36 | 0.65 |
| AlphaFold [31] | **0.56** | **0.76** |

(a) **Backbone on S669.** MSA inputs improve performance. A pre-trained AlphaFold model performs the strongest.

| aggregator | nDCG | DetPr | AUC |
|---|---|---|---|
| outer prod. + flat. | 0.39 | 0.15 | 0.82 |
| product | 0.42 | 0.14 | **0.85** |
| sum | **0.43** | **0.16** | 0.84 |
| whiten + sum | 0.25 | 0.12 | 0.80 |

(b) **Mutation Aggregation on cDNA2.** Higher-order mutations aggregate single mutation features. Adding these features is competitive.

| backbone opt | $r_s$ | AUC |
|---|---|---|
| freeze | 0.48 | 0.72 |
| finetune | **0.56** | **0.76** |

(c) **Backbone Optimization on S669.** Fine-tuning the AlphaFold backbone improves performance.

| higher model | nDCG | DetPr | AUC |
|---|---|---|---|
| direct | 0.40 | 0.13 | **0.85** |
| multiply | 0.17 | 0.08 | 0.77 |
| add | **0.43** | **0.16** | 0.84 |

(d) **Higher-order Modeling on cDNA2.** We model the $\Delta\Delta G$ of the double mutation as a residual combined with the $\Delta\Delta G$ of its single mutation.

Table 5: *Mutate Everything* ablation experiments. Default Settings are marked in grey.

**Backbone Optimization** We try fine-tuning and freezing the AlphaFold backbone during stability fine-tuning in Table 5c. We find that fine-tuning the backbone improves stability prediction performance.

**Aggregation** The technique to aggregate mutation representations for higher-order mutations is ablated in Table 5b. To control for feature dimension size, we reduce the head dimension when aggregating with outer product and flattening. Aggregating with product and summation performs similarly on double mutations. We add the features for a natural extension to more mutations.

**Higher Order Modeling** In the direct prediction model, the multi-mutant head directly predicts $\Delta\Delta G$. In the multiply and add models, the multi-mutant head learns how interactions among single mutations affect the higher-order mutation's $\Delta\Delta G$. In these models, the multi-mutant head output learns an additive bias or multiplicative scaling to the sum of single mutation $\Delta\Delta G$s. In Table 5d, we show that learning a bias performs the strongest among these three options.

# 6 Limitations

First, our training dataset contains biases that may affect model performance. The training set contains only small proteins, which may limit performance on larger ones. Our model may exhibit biases towards certain types of mutations due to the data imbalance in our training set. Second, the limited availability of experimental stability data poses a challenge for in-silico evaluation. Evaluation on larger and more diverse datasets is necessary to fully assess the generalizability of our model. In the future, we hope that high-throughput experimental assays will enable more rigorous evaluation and further improvements in protein stability prediction.

# 7 Conclusion

We present a method that efficiently scales thermodynamic stability prediction from single mutations to higher-order mutations. Our key insight is that the effects of mutations on the same protein are correlated. Thus, for a target protein, it suffices to run a deep backbone once and decode the effect of all mutations simultaneously using a shallow decoder. With the AlphaFold model as our backbone, our method outperforms existing methods on a variety of single and multiple mutation benchmarks. Our method scales to millions of mutations with minimal computational overhead and runs in a fraction of the time it would take prior works.

# 8 Acknowledgements

This work is supported by the NSF AI Institute for Foundations of Machine Learning (IFML).

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

# A  Experimental Setting

We elaborate on our experimental protocol, including the datasets used for training and testing, the evaluation metrics, and the baselines that calibrate our models.

## A.1  Datasets

A fundamental principle in machine learning is to partition data so that the model is not evaluated on the same data it was trained on. This is essential to assess the model's ability to generalize beyond its training data (in our case, to new proteins). Protein sequences are regarded as the same protein when they have large sequence similarity (e.g. $> 50\%$), even when they are not exactly the same string. The impact of training on protein sequences similar to those in the test set has received considerable attention [18, 45, 54]. We list the datasets we use below and report the maximum sequence similarity of our training set with each test set. This ensures a clear separation between our training and test sets.

**cDNA proteolysis** [72] is a large scale dataset containing mutant proteins with $\Delta\Delta G$ measurements. The dataset is generated by cDNA display proteolysis, a massively parallel technique that measures thermodynamic folding stability for millions of protein variants [72]. The cDNA proteins average 56.1 amino acids in length with a maximum length of 72 and a minimum length of 30 amino acids. The mean MSA depth is 7797 with a standard deviation of 6282. The maximum depth is 23525 and the minimum depth is 5. The mutations contain either one or two substitutions on a small protein. We follow the train-val split introduced in [18] and additionally filter out proteins in our training set that are similar to those in our evaluation datasets. Specifically, we train on 116 proteins with 213,000 total mutations, of which 97,000 are double mutants and 117,000 are single mutants. We hold out cDNA2, a validation set of 18 mini-proteins with 22,000 total double mutations. The proteins in our training set have at most 36% sequence similarity to those in cDNA2. A comprehensive analysis of the dataset, experimental assay, and filtering criteria are found in their paper [1].

**ProTherm** [48] contains experimentally measured thermodynamic data $\Delta\Delta G$ for protein mutants. We use PTMul, a split containing only higher-order mutations curated by DDGun [44]. There are 88 proteins with 846 total mutations in PTMul. The proteins in our training set have at most 35% sequence similarity to those in PTMul.

**S669** [54] is a manually curated dataset of mutant proteins and their $\Delta\Delta G$ measurements. S669 contains mutations with one substitution. There are 94 proteins with 669 total single mutations in S669. The proteins in our training set have at most 30% sequence similarity to those in S669.

**ProteinGym** [49] is a curated set of proteins in which mutations are assessed for some fitness phenotype. These fitness phenotypes vary drastically, from cell growth to expression to fluorescence. The fitness values are always normalized so a higher value indicates stronger fitness. For classification metrics, the fitness values are binarized by some threshold either defined manually or arbitrarily at the median value. We also consider a subset of the proteins in ProteinGym in which fitness correlates highly with thermodynamic stability. These phenotypes include thermostability, expression, etc. There are 87 proteins totaling over 1.6 million (potentially higher-order) mutations. The proteins in our training set have at most 45% sequence similarity to those in ProteinGym.

**T2837** [18] is an aggregate dataset with mutant proteins and their $\Delta\Delta G$ measurements. T2837 contains mutations with one substitution. There are 129 proteins with 2837 total single mutations in T2837. The proteins in our training set have at most 34% sequence similarity to those in T2837.

## A.2  Metrics

We evaluate our model's ability to assess mutations in ProTherm, S669, and ProteinGym. Additionally, we evaluate our model's ability to detect stabilizing mutations in cDNA2.

### A.2.1  Mutation Assessment

The model predicts the $\Delta\Delta G$ measurement under all mutations with experimental data. The models are evaluated by the following regression and classification metrics. The classification metrics are

useful in applications where it is unnecessary to provide exact $\Delta\Delta G$ and a binary stabilizing or de-stabilizing label (or third label, neutral) is sufficient.

Following the literature, we additionally evaluate reverse mutations (in contrast to the standard "forward" mutations) where the "to" and "from" amino acids are flipped (e.g. P1K to K1P). The model takes the mutant sequence as input and is expected to output the negation of the original $\Delta\Delta G$ value when mutating to the original native amino acid.

**Spearman correlation coefficient ($\mathbf{r}_s$)** evaluates the model's ability to rank the mutations by their $\Delta\Delta G$ values. This metric considers only the relative ordering between mutations and disregards the predicted values. Spearman has its limitations because our datasets contain predominantly destabilizing mutations ($\Delta\Delta G > 0$). Spearman applied on these datasets overwhelmingly measures the model's ability to rank destabilizing mutations which does not directly translate to identifying the most stabilizing mutations [8, 58].

**Root Mean Square Error (RMSE)** provides a measure of how closely the predicted measurements align with the true measurements.

**Area Under the receiver operating characteristic Curve (AUC)** is a popular classification metric that focuses on positives. The receiver operating characteristic curve is created by plotting precision against recall at various classification thresholds. Precision is calculated as the ratio of true positives to the sum of true positives and false positives. Recall is calculated as the ratio of true positives to the sum of true positives and false negatives.

**Matthew correlation coefficient (MCC)** evaluates a model's classification performance under imbalanced datasets. It takes into account true positive, true negative, false positive, and false negative predictions to measure the overall performance of a model.

### A.2.2   Mutation Detection

The regression metrics for mutation assessment are not directly suitable for protein engineering where the goal is to find stabilizing mutations. In addition to the classification metrics, we evaluate our models using *per-protein* metrics which evaluate the model's ability to detect a few stabilizing mutations from a large number of possible mutations. These metrics are applied to heavily annotated proteins ($\sim 1000$ $\Delta\Delta G$ measurements) with randomly sampled mutations.

**Stabilizing Spearman correlation coefficient $\mathbf{r}_s$** is the Spearman coefficient calculated on the subset of experimentally stabilizing mutations. This mimics the case in protein engineering where there are too many stabilizing mutations to test in vitro and the goal is to prioritize the most stabilizing ones. This metric is used in [18, 65].

**Normalized Discounted Cumulative Gain** (nDCG) [30] quantifies the quality of a ranking among mutations by considering both the experimental $\Delta\Delta G$ and predicted rank of each mutation. Specifically, nDCG computes the sum of experimentally stabilizing $\Delta\Delta G$s weighted inversely by their ranks. This metric rewards a model that ranks the best mutations first. We truncate the ranking at $K = 30$ mutations to approximate the number of in-vitro experiments that can typically be run in parallel. This metric is found in [59].

**Detection Precision (DetPr)** measures the fraction of experimentally stabilizing mutations among the top $K = 30$ predicted mutations by the model. This simulates the success rate a protein engineer might expect when deploying this model. $K$ is set to approximate the number of in-vitro experiments that can typically be run in parallel.

### A.3   Baselines

Several baselines calibrate performance across datasets. A subset of the critical baselines is introduced briefly (see [54] for more).

**Mean** is a simple baseline in which the mean measurement is predicted for all mutations. This baseline is not conditioned on the protein or mutation and instead uses only the dataset statistic.

**Multiple Sequence Alignment (MSA)** leverages the statistics from a set of similar protein sequences to rank mutations. Similar protein sequences are fetched, aligned, and indexed at a mutation position.

This yields an amino acid frequency count normalized into a probability distribution. The score is defined as the likelihood ratio between the mutated and native amino acid types. While calibrated for a given position and protein, it may not be well-calibrated across proteins.

**ESM [38]** also computes the likelihood ratio between a mutated and native amino acid type [42]. Unlike MSA, the probability distribution is decoded from ESM2 [38], a model pre-trained on masked amino acid prediction. Like MSA, ESM may not be well-calibrated across proteins.

**DDGun [44]** uses a linear combination of scores derived from evolutionary and structural features to compute $\Delta\Delta G$.

**FoldX [67]** leverages an empirical force field to estimate the change in stability. It models the Van Der Waals interactions, solvation and hydrogen bonding interactions, changes in entropy, and other physical values.

**PROSTATA [74]** is a sequence model for stability prediction built on the ESM2 model. It takes native and mutant protein sequences as input and produces a $\Delta\Delta G$ value.

**Stability Oracle [18]** is a structure-based graph-transformer model for stability prediction.

**Tranception [49]** is a specialized autoregressive transformer that uses grouped attention to encourage specialization across heads.

# B    Additional Results

## B.1    Higher-order Mutation Results on PTMul Breakdown

ProTherm Multiple (PTMul) contains 846 mutations, of which 536 are double mutations and 310 mutations with more than two mutations. We breakdown the performance on PTMul into double mutations and higher-order mutations (greater than 2) in Table 6. While our model performs similarly against the additive baseline on double mutations, *Mutate Everything* outperforms the additive baseline on higher-order mutations.

| Method | Double Mutations | | | | More Mutations | | | |
|---|---|---|---|---|---|---|---|---|
| | $r_s$ | AUC | MCC | RMSE$_\downarrow$ | $r_s$ | AUC | MCC | RMSE$_\downarrow$ |
| DDGun | 0.28 | 0.63 | 0.22 | 2.23 | 0.15 | 0.57 | 0.12 | 2.19 |
| DDGun3D | 0.29 | 0.61 | 0.17 | 2.25 | 0.19 | 0.60 | 0.17 | 2.20 |
| Additive Baseline | **0.52** | **0.76** | 0.30 | **1.95** | 0.49 | 0.79 | 0.45 | 2.14 |
| *Mutate Everything* (ours) | 0.50 | 0.75 | **0.34** | 2.08 | **0.60** | **0.86** | **0.58** | **1.99** |

Table 6: **Results on PTMul splits containing double mutations and more than 2 mutations.** *Mutate Everything* outperforms existing methods, especially when there are multiple mutations.

| Method | Forward | | | | Reverse | | | |
|---|---|---|---|---|---|---|---|---|
| | $r_s$ | AUC | MCC | RMSE $_\downarrow$ | $r_s$ | AUC | MCC | RMSE $_\downarrow$ |
| PROSTATA-IFML [18, 74] | 0.53 | 0.75 | 0.31 | 1.75 | 0.53 | 0.75 | 0.32 | 1.75 |
| Stability Oracle [18] | 0.62 | **0.81** | **0.39** | 1.65 | **0.62** | **0.81** | **0.39** | **1.65** |
| *Mutate Everything* (Ours) | **0.65** | 0.79 | **0.39** | **1.59** | 0.52 | 0.72 | 0.28 | 1.75 |

Table 7: **Comparisons on Single Mutation Thermodynamic Stability prediction in T2837.** We report regression and classification metrics on the forward and reverse datasets. $r_s$ refers to Spearman correlation coefficient. Our training proteins have at most 34% sequence similarity with those in T2837. *Mutate Everything* is state-of-the-art on T2837 forward.

| Test Set | #Mutations | $r_p$ | $r_s$ | RMSE $_\downarrow$ | MCC | AUROC | Precision | Recall | Accuracy |
|---|---|---|---|---|---|---|---|---|---|
| T2837 Orig | 2835 | 0.6274 | 0.6472 | 1.5877 | 0.3888 | 0.7956 | 0.5615 | 0.4791 | 0.7894 |
| T2837 TR | 2835 | 0.5301 | 0.5208 | 1.7563 | 0.2802 | 0.7229 | 0.8297 | 0.8572 | 0.7517 |
| T2837 Orig + TR | 5670 | 0.7229 | 0.7282 | 1.6741 | 0.5482 | 0.8433 | 0.7765 | 0.7701 | 0.7705 |

Table 8: *Mutate Everything* **regression and classification metrics on T2837 and its reverse datasets.** TR refers to Thermodynamic Reversibility. $r_p$ is Pearson correlation coefficient. $r_s$ is Spearman correlation coefficient. Mutations that failed our data pipeline are excluded.

## B.2 Single Mutation Results on T2837

We include our results on the newly proposed T2837 dataset in Table 7 [18]. *Mutate Everything* achieves a Spearman correlation of 0.65 compared to 0.62 of Stability Oracle, outperforming existing methods on the forward dataset. PROSTATA-IFML [18] fine-tunes PROSTATA [74] on the cDNA proteolysis dataset for more direct comparison. *Mutate Everything* performs competitively with other sequence models on the reverse dataset even without training on reverse mutations. We show our performance on additional metrics for T2837 in Table 8.

