# OpenReview forum: "Predicting a Protein's Stability under a Million Mutations"
_NeurIPS.cc/2023/Conference — NeurIPS 2023 poster_

### Official Review · Reviewer_31pU · 2023-06-28

**Soundness:** 4 excellent
**Presentation:** 3 good
**Contribution:** 3 good
**Rating:** 8
**Confidence:** 4

**Summary:**

The paper proposes a new method, Mutate Everything Method, to model a protein's thermodynamic stability based on mutations in a protein's sequence. A key distinguishing feature of the method is the ability to perform large amount of parallel evaluations, which significantly speed up computational efficiency. The authors first introduce the general challenge in designing stable protein though mutations, describe thermal stability as a physical metric and introduce the general outline of their method. Next, the authors describe related work in protein engineering, protein structure prediction and protein stability modeling followed by a more precise, mathematical description of the problem setup for the proposed method. Essentially, the Mutate Everything Method relies on taking protein sequence embeddings from pretrained models (AlphaFold, ESM) and then predicting thermal stability of the mutations with lightweight MLP heads with one head corresponding to each mutation. In the case of multiple mutations, the authors aggregate the outputs from each individual mutation head. In their experiments, the authors study their method on a variety of sequence mutation datasets, including both single and higher-order mutations with the general results indicating better modeling and compute performance of the Mutate Everything Method. The order also perform an ablation of representations from different embedding encoders, including AlphaFold, ESM2 and MSA-Transformer.

**Strengths:**

The paper has the following strengths:
* Originality: The paper proposes a new, pragmatic method for a relevant problem in modeling how mutations affect a protein thermodynamic stability.
* Quality: The paper performs detailed evaluations of the method on multiple types of tasks, including multiple datasets, and compares to various other methods along both modeling performance and compute efficiency.
* Clarity: The purpose, goals and details of the method and results are generally well presented.
* Significance: The paper tackles a relevant problem and show notable performance improvements in multiple settings.

**Weaknesses:**

The paper could be improved by:
* Providing more clarity into the parallel evaluation pipeline. Figure 3 is not very clear in how exactly multiple mutation evaluations are processed and aggregated. I suggest adding labels that show how a mutation changes a particular part in the sequence similar to Figure 2. [Clarity]
* The analysis on homology in Section 5.3 is interesting and adds an important dimension on how to properly design data splits for these kinds of tasks. It would be nice to see a deeper discussion on this; potentially in the appendix due to space constraints. [Quality, Significance]



**Questions:**

* Can you provide more detail on how you ensure the mapping of the mutation in the embedding space is aligned with the mapping of the mutation in the original sequence space?
* Can you describe in more detail how your parallel evaluation ensure proper mapping of mutations to the correct protein sequence? I am trying to get a better sense of whether the parallel evaluation are essentially operating by creating a larger "meta-sequence" or if there is any type of multi-processing happening.
* It seems like the method requires a frozen embedding encoder as it is implemented right now. How robust would the method be to finetuning the encoder?

**Limitations:**

The authors include a section on limitations mainly focused on modeling shortcomings.

---

> ### Author Rebuttal · Authors · 2023-08-10
>
> > Figure 3 is not very clear in how exactly multiple mutation evaluations are processed and aggregated. I suggest adding labels that show how a mutation changes a particular part in the sequence similar to Figure 2. [Clarity]
>
> Figure 3 showcases how we can apply Mutate Everything to decode ΔΔG values in parallel. How the multiple mutation head processes and aggregates is identical to that shown in Figure 2c. We will clarify the head in Figure 3.
>
> > The analysis on homology in Section 5.3 is interesting and adds an important dimension on how to properly design data splits for these kinds of tasks. It would be nice to see a deeper discussion on this; potentially in the appendix due to space constraints. [Quality, Significance]
>
> We agree that homology plays an essential role in designing data splits. There are many works in the literature that recognize this problem [1] and design more informed dataset splits [2,3,4]. We will elaborate on this discussion in the additional page provided in the final version.
>
> > Can you provide more detail on how you ensure the mapping of the mutation in the embedding space is aligned with the mapping of the mutation in the original sequence space?
>
> Our training objective aligns the embeddings with the input sequence. We have an embedding F[L,t] for each position L and “to” amino acid t. The loss aligns a mutation at position L to amino acid t (and its ΔΔG) with the corresponding embedding F[L,t].
>
> > Can you describe in more detail how your parallel evaluation ensure proper mapping of mutations to the correct protein sequence? I am trying to get a better sense of whether the parallel evaluation are essentially operating by creating a larger "meta-sequence" or if there is any type of multi-processing happening.
>
> Single mutations are modeled exhaustively. The decoder outputs a set of Lx20 single mutation representations from which all single mutation ΔΔGs are decoded. For example, F[10,A] decodes the change in stability when position 10 is mutated to Alanine. During training, the mutation representation is aligned to the experimental ground truth ΔΔG.
>
> All higher-order mutations are made up of the same set of Lx20 single mutations. In our model, these single mutation representations are indexed and aggregated to represent higher-order mutations. For example, a mutation at position 10 to Alanine and 15 to Cysteine is represented as F[10,A] + F[15,C]. The loss will ensure alignment to experimental ΔΔG.
>
> Computationally, we perform one forward pass through the backbone network. The backbone is the computationally most expensive portion of the network. Computation does not depend on the number of higher order mutations considered. ΔΔGs of higher order mutations are computed as inner sums between embeddings at certain positions, and are thus computationally very fast. They do however scale exponentially with the order of mutation considered (there is a linear number of single point mutations, a quadratic number of two point mutations, a cubic number of three point mutations, …).
>
> The structure of our model enables us to compute millions of ΔΔG values for one protein with only a single forward of the backbone.
>
> > It seems like the method requires a frozen embedding encoder as it is implemented right now. How robust would the method be to finetuning the encoder?
>
> We find that fine-tuning the AlphaFold2 backbone improves performance. The current method fine-tunes the backbone.
> |           | Spearman | AUC  | MCC  | RMSE |
> |-----------|----------|------|------|------|
> | Freeze    | 0.48     | 0.72 | 0.20 | 1.46 |
> | Fine-tune | 0.56     | 0.76 | 0.37 | 1.36 |
>
>
> [1] Montanucci L, Savojardo C, Martelli PL, Casadio R, Fariselli P. On the biases in predictions of protein stability changes upon variations: the INPS test case.
>
> [2] Li, B., Yang, Y.T., Capra, J.A., Gerstein, M.B.: Predicting changes in protein thermodynamic stability upon point mutation with deep 3d convolutional neural networks.
>
> [3] Pancotti, C., Benevenuta, S., Birolo, G., Alberini, V., Repetto, V., Sanavia, T., Capriotti, T. and Fariselli, P. Predicting protein stability changes upon single-point mutation: a thorough comparison of the available tools on a new dataset.
>
> [4] Diaz, D.J., Gong, C., Ouyang-Zhang, J., Loy, J.M., Wells, J.T., Yang, D., Ellington, A.J., Dimakis, A., Klivans, A.R.: Stability oracle: A structure-based graph-transformer for identifying stabilizing mutations.

---

> > ### Comment · Reviewer_31pU · 2023-08-14
> > **Thank you for additional details**
> >
> > Thank you for the additional details. Most of the questions and concern have been addressed.

---

### Official Review · Reviewer_xRxg · 2023-06-30

**Soundness:** 3 good
**Presentation:** 3 good
**Contribution:** 2 fair
**Rating:** 5
**Confidence:** 4

**Summary:**

The authors present the "Mutate Everything Method", a simple method that builds on top of protein representations obtained through existing models to predict the effect of single and higher-order mutations. They apply the method to representations from ESM2 and AlphaFold, and show the performance on several downstream tasks.

**Strengths:**

The proposed method is based on a very simple but effective idea, and the paper is written in a clear and mostly comprehensive manner. This seems like a relatively low-resource and fast method to get more use out of existing representations. Additionally, a substantial amount of results is presented over a wide range of benchmarks and metrics.

**Weaknesses:**

Even though I enjoyed reading this paper and I appreciate the idea that a simple method can make a difference for real downstream tasks, I am not entirely convinced that this method, which esentially consists of training a lot of MLPs on top of existing representations, fits the NeurIPS venue well. It doesn't seem like a substantial contribution to the field of machine learning, and this paper might therefore be better suited for a different venue where it would have more impact. Moreover, the reported spearman correlations to experimental results are very low in some places, and around 0.5 at most, signifying a very weak correlation at best. This begs the question whether this is an unfortunate metric or if there's a deeper issue causing the correlation to experimental results to be weak.

**Questions:**

1. The way the method is presented, it seems like the authors claim that it will be possible to reliably predict the effect of single mutations as well as high-order effects. Even though some evidence is presented, it is not conclusive, especially not for the higher-order case. I understand that this depends on e.g. the availability of data sets, but perhaps the presentation can be a bit more nuanced.
2. Even though the method is presented as being simple, there seem to be a lot of MLPs to train! Can you give more details about these MLPs (i.e. how many layers, size of the layers) and the overall memory requirements?
3. The paper goes straight from results to "Limitations", without any conclusions or discussion. As a reader, this section is clearly missing.
4. Table 4, 5, and 6 show no standard deviations. Is there a good reason for that? Otherwise, it would be great if those could be added.
5. The part where you predict the residual between single mutation $\Delta\Delta G$s and teh combined effect is a bit vague to me. Why was this beneficial?
6. Related to the previous point, when you say in line 174-175 that you learn the residual to the sum of experimental $\Delta\Delta G$s for the constituent single mutations, what does this mean? Are you giving the model the experimental values of single mutations? Does this mean you always need to have experimental data for single effects if you want to predict combined effects? Or am I misunderstanding the sentence? The wording is a bit confusing.
7. For clarity, it would be great if you could incorporate $x$, $\mu$, and $z$ (and perhaps $f$ and $h$) in Figure 2.
8. What is the dimensionality of $d$?
9. Some of the baselines only show up in tables without really being explained in the main text or the appendix.
10. There seem to be some references missing in the first part of the introduction (line 25-30), about the use of machine learning for finding stabilizing mutations. And perhaps more citations can be added to section 2.3 (line 104-111) as well.

**Limitations:**

The authors describe the limitations of their work briefly but accurately. The only thing missing could be a discussion about the low spearman correlation values (in general).

---

> ### Author Rebuttal · Authors · 2023-08-09
>
> > Reported spearman correlations to experimental results are very low in some places, and around 0.5 at most, signifying a very weak correlation at best. This begs the question whether this is an unfortunate metric or if there's a deeper issue causing the correlation to experimental results to be weak.
>
> We agree that the spearman correlation coefficient can be a suboptimal metric. Many datasets contain predominantly destabilizing mutations. Spearman overwhelmingly measures the model’s ability to rank destabilizing mutations [2,3], which does not directly translate to identifying the most stabilizing mutations.
>
> Prior works have proposed using metrics more in line with finding stabilizing mutations. Area under precision-recall curve (AUC), Matthews correlation coefficient (MCC) [1], and normalized Discounted Cumulative Gain (nDCG) [4] assess a model’s ability to identify stabilizing mutations. We report all metrics including Spearman for completeness.
> We will include a discussion of the metrics in the final version.
>
> > The way the method is presented, it seems like the authors claim that it will be possible to reliably predict the effect of single mutations as well as high-order effects. Even though some evidence is presented, it is not conclusive, especially not for the higher-order case. I understand that this depends on e.g. the availability of data sets, but perhaps the presentation can be a bit more nuanced.
>
> Rigorously studying higher-order mutations is hard. Public data is scarce and expensive to collect. The data that does exist does support our conclusions. We express our concerns in the Limitations section in line 285. We are more than happy to add nuance to the presentation where needed. What were the things the reviewer would like to change?
>
> > MLP details and memory requirements
>
> An adapter maps the backbone-specific hidden dimensionality to D=128 and all subsequent layers operate at D=128.
>
> Each amino acid projection $f^t$ in the amino acid expansion is a linear layer.
>
> The single mutation decoder $g^1$ is a linear layer.
>
> The higher-order mutation decoder transforms the previous embedding with a 2-layer MLP. These representations are aggregated and are fed into a 3-layer MLP to predict ddg.
> The MLPs use LayerNorm and ReLU.
>
> We train on single and double mutations from small proteins with at most 72 amino acids. This keeps the memory requirement small for the higher-order decoder.
>
> Thank you for the feedback, we will report these thoroughly in the paper and release code upon acceptance.
>
> > Missing Conclusion
>
> We omitted a conclusion due to space constraints. We will include one in the additional page allotted for the final version.
>
> > Table 4, 5, and 6 show no standard deviations. Is there a good reason for that? Otherwise, it would be great if those could be added.
>
> Thank you for the suggestion. We will add standard errors for all of our experiments in the final version.
>
> > The part where you predict the residual between single mutation ΔΔGs and the combined effect is a bit vague to me. Why was this beneficial?
>
> We tried predicting the residual and directly predicting the combined ΔΔG for higher order mutations. We found predicting the residual to be easier to train. We will mention this in the exposition.
>
> > Related to the previous point, when you say in line 174-175 that you learn the residual to the sum of experimental  ΔΔGs for the constituent single mutations, what does this mean? Are you giving the model the experimental values of single mutations? Does this mean you always need to have experimental data for single effects if you want to predict combined effects?
>
> We do not provide the model with any experimental ΔΔG values. For higher order mutations, our model first predicts the ΔΔG for each constituent single mutation (computationally), then predicts a residual to the sum.
>
> > For clarity, it would be great if you could incorporate x, μ, and  z  (and perhaps f and h) in Figure 2.
>
> Great idea, thank you. We will add the symbols to the figure.
>
> > What is the dimensionality of d?
>
> We map a backbone-specific hidden dimension to D=128.
>
> > Some of the baselines only show up in tables without really being explained in the main text or the appendix.
>
> We explain our baselines in Supplement Section 1.3 and provide citations for existing works. We will try to work this into the final version given the extra content page.
>
> > There seem to be some references missing in the first part of the introduction (line 25-30), about the use of machine learning for finding stabilizing mutations. And perhaps more citations can be added to section 2.3(line 104-111) as well.
>
> Thank you. Will do. One citation for the explosion in biological data is “Mega-scale experimental analysis of protein folding stability in biology and protein design” by Kotaro Tsuboyama, Justas Dauparas, Jonathan Chen, Niall M. Mangan, Sergey Ovchinnikov, Gabriel J. Rocklin.
>
> [1] Broom, A., Trainor, K., Jacobi, Z., Meiering, E.M. Computational Modeling of Protein Stability: Quantitative Analysis Reveals Solutions to Pervasive Problems.
>
> [2]  Benevenuta, S., Birolo, G., Sanavia, T., Capriotti, E., Fariselli, P. Challenges in predicting stabilizing variations: An exploration.
>
> [3] Pucci, F., Schwersensky, M., Rooman, M. Artificial intelligence challenges for predicting the impact of mutations on protein stability.
>
> [4] Qiu, Y., Wei, G.W. Persistent spectral theory-guided protein engineering.

---

> > ### Comment · Reviewer_xRxg · 2023-08-15
> >
> > I would like to thank the authors for their detailed reply. Here's my response:
> >
> > * **Venue)** I noticed you did not respond to whether or not your paper suits the NeurIPS venue. Just to clarify, this comment was really not meant to attack your paper, as I would gladly be convinced that this paper matches the venue well. Can you comment on this?
> > * **Metrics)** Okay, fair enough. I still think these Spearman correlations are so low that they’re barely worth reporting, but I understand that you’re following existing literature for this metric. Thank you for clarifying.
> > * **Higher-order mutations)** Examples could be line 9-10 of the Abstract, or line 37-42 of the Introduction. Again, I fully appreciate how hard (or perhaps currently impossible) it is to rigorously benchmark performance for high-order mutations. I am merely suggesting to provide a bit of a disclaimer for this specific part of your method. It is of great added value that your method can handle higher-order mutations, but no conclusive evidence can be given for its reliability in this setting.
> > * **Model and memory details)** Thank you for the detailed answer.
> > * **Missing conclusion)** Okay.
> > * **Standard deviations)** What was the reason behind the decision to not include standard deviations, neither in the original manuscript nor in the rebuttal period where a pdf with one extra page of results could be provided? Is this only due to time constraints or is there some other reason?
> > * **Predicting the residual)** Can you give any intuition on why predicting the residual is helpful, or is it purely an empirical observation?
> > * **Line 174-175)** Ah I see, so “experimental” here actually means predicted by the model? If so, then I would recommend rewording this sentence a bit for clarity.
> > * **Figure 2)** Great, thanks.
> > * **Dimensionality of $d$ )** Thank you, please include this in the paper/appendix if it’s indeed not there yet.
> > * **Baseline explanations)** *Mean*, *MSA*, and *ESM* are indeed described in the appendix, but for the general reader, baseline methods like DeepSequence and EVE might require a brief explanation/motivation (so for example “VAE-based method with a Bayesian decoder” or something similar).
> > * **References)** Thank you for adding more references.

---

> > > ### Author Response · Authors · 2023-08-17
> > > **paper suits the NeurIPS venue**
> > >
> > > >  [Reviewer] Even though I enjoyed reading this paper and I appreciate the idea that a simple method can make a difference for real downstream tasks, I am not entirely convinced that this method, which essentially consists of training a lot of MLPs on top of existing representations, fits the NeurIPS venue well. It doesn't seem like a substantial contribution to the field of machine learning, and this paper might therefore be better suited for a different venue where it would have more impact.
> > >
> > > >> [Reviewer] Venue) I noticed you did not respond to whether or not your paper suits the NeurIPS venue. Just to clarify, this comment was really not meant to attack your paper, as I would gladly be convinced that this paper matches the venue well. Can you comment on this?
> > >
> > > According to the call for papers, NeurIPS welcomes research in machine learning for science. The ML community introduced some of the most exciting benchmarks relevant to our method and we have rigorously evaluated our models against it [1]. This highlights the community’s interest in the problems we address. We initially did not respond to this comment because we believe the scope of the conference is better defined at ACs, SACs, and PCs. If scope is a true concern it might be worth escalating this discussion, since ACs, SACs, and PCs are the ultimate authority on scope.
> > >
> > > We obviously would argue to include more scientific (and biologically grounded) research. NeurIPS has always been a very inclusive venue, which many consider its core strength. NeurIPS has always undergone tremendous change, and always included and absorbed new fields. In the 90s, old-school computational neuroscience made up a large part of NeurIPS. In the 2000s, classical machine learning found a home at NeurIPS, while Neural Networks were shunned. In the early 2010s, NeurIPS was among the first venues to welcome deep learning back. As for the exact research topic at hand, as far back as the second NIPS 1989 research on protein sequences appeared at this venue [2]. Bengio et al [2] adapted their own speech recognition system [3] to detect homologies in proteins. In fact, NeurIPS features a paper studying protein structures more years that it didn’t.
> > >
> > > [1] Notin, P., Dias, M., Frazer, J., Hurtado, J.M., Gomez, A.N., Marks, D., Gal, Y.: Tranception: protein fitness prediction with autoregressive transformers and inference-time retrieval. ICML 2022.
> > >
> > > [2] Bengio, Y., Bengio, S., Pouliot, T., Agin, P. A Neural Network to Detect Homologies in Proteins. NIPS 1989.
> > >
> > > [3] Bengio, Y., Cardin, R., De Mori, R., Merlo, E.M. Programmable execution of multi-layered networks for automatic speech recognition. Communications of the ACM 1989.

---

> > > > ### Author Response · Authors · 2023-08-17
> > > > **Thank you for the discussion.**
> > > >
> > > >
> > > > > Higher-order mutations) Examples could be line 9-10 of the Abstract, or line 37-42 of the Introduction. Again, I fully appreciate how hard (or perhaps currently impossible) it is to rigorously benchmark performance for high-order mutations. I am merely suggesting to provide a bit of a disclaimer for this specific part of your method. It is of great added value that your method can handle higher-order mutations, but no conclusive evidence can be given for its reliability in this setting.
> > > >
> > > > Thank you. We will definitely rephrase these parts for the final version.
> > > >
> > > > > [Reviewer] Table 4, 5, and 6 show no standard deviations. Is there a good reason for that? Otherwise, it would be great if those could be added.
> > > > >> [Author] Thank you for the suggestion. We will add standard errors for all of our experiments in the final version.
> > > > >>> [Reviewer] What was the reason behind the decision to not include standard deviations, neither in the original manuscript nor in the rebuttal period where a pdf with one extra page of results could be provided? Is this only due to time constraints or is there some other reason?
> > > >
> > > > Here are our results on S669 forward with standard errors for Table 4 over 6 runs.
> > > >
> > > > | Spearman    | AUROC       | MCC         | RMSE        |
> > > > |-------------|-------------|-------------|-------------|
> > > > | 0.56 (0.01) | 0.75 (0.01) | 0.34 (0.01) | 1.38 (0.01) |
> > > >
> > > > Here are our results on ProteinGym-Stability with standard errors for Table 5 over 6 runs.
> > > > | Spearman    | AUROC       | MCC         |
> > > > |-------------|-------------|-------------|
> > > > | 0.53 (0.01) | 0.78 (0.01) | 0.41 (0.01) |
> > > >
> > > > Here are our results on ProteinGym with standard errors for Table 5 over 6 runs.
> > > > | Spearman    | AUROC       | MCC         |
> > > > |-------------|-------------|-------------|
> > > > | 0.48 (0.01) | 0.77 (0.01) | 0.38 (0.01) |
> > > >
> > > >
> > > > We will add standard errors for all of our other experiments in the final version.
> > > >
> > > >
> > > > > Predicting the residual) Can you give any intuition on why predicting the residual is helpful, or is it purely an empirical observation?
> > > >
> > > > This is an empirical observation. Intuitively, it may be easier for the model to learn the residual rather than the original function (See ResNets [1]).
> > > >
> > > > > Line 174-175) Ah I see, so “experimental” here actually means predicted by the model? If so, then I would recommend rewording this sentence a bit for clarity.
> > > >
> > > > Yes, thank you for the recommendation. We will revise L174-175 to “computationally predicted ΔΔG” instead of “experimental ΔΔG”
> > > >
> > > > > Baseline explanations) Mean, MSA, and ESM are indeed described in the appendix, but for the general reader, baseline methods like DeepSequence and EVE might require a brief explanation/motivation (so for example “VAE-based method with a Bayesian decoder” or something similar).
> > > >
> > > > Thank you for the recommendation. We will add brief descriptions for the baselines to the appendix.
> > > >
> > > > [1] He, K., Zhang, X., Ren, S., Sun, J. "Deep residual learning for image recognition."

---

> > > > > ### Comment · Reviewer_xRxg · 2023-08-17
> > > > >
> > > > > * **Standard deviations**
> > > > >
> > > > >     Thank you for providing these results, I’m happy to see the performance is so consistent.
> > > > >
> > > > > * **Other comments**
> > > > >
> > > > >     All further concerns we addressed accurately.
> > > > >
> > > > >
> > > > > * **Conclusion**
> > > > >
> > > > >     For now, I will not raise my score due to my concerns about novelty (see previous reply). However, if later discussion alleviates those concerns, I could be on board with accepting this paper.

---

> > > > ### Comment · Reviewer_xRxg · 2023-08-17
> > > > **Venue**
> > > >
> > > > Let me be very clear that of course I agree that machine learning for science is very welcome at NeurIPS, and this conference features several interesting applied ML papers each year. However, I still feel this paper might fall short on the innovation front, as I’m not sure if the contributions are novel enough, neither in the field of machine learning nor in the field of protein modelling. But, like you said, this is up for further discussion. Thank you for your perspective.

---

### Official Review · Reviewer_AB6d · 2023-07-05

**Soundness:** 3 good
**Presentation:** 4 excellent
**Contribution:** 3 good
**Rating:** 7
**Confidence:** 4

**Summary:**

The authors are concerned with the task of predicting the effects of single- and double-residue mutations on the thermodynamic stability of a protein. They propose a simple method that involves passing combinations of embeddings from a pretrained model (AlphaFold2 or ESM) to MLPs. Compared to existing approaches, theirs is computationally efficient and conceptually simple. It also achieves good results on various benchmarks.

**Strengths:**

The method is original, efficient, simple to understand, and works well. It addresses an important question in structural biology that is currently far from a solution. It has several favorable properties; for example, one can hot-swap in better embeddings as new models become available.

**Weaknesses:**

Some of the evaluations lack some important details, and a few claims seem questionable to me. See the "questions" section below. Overall, I think the evaluation for high-order mutations is a little light. In particular, while the model is evaluated on a mixture of double- and triple-residue mutations, individual results for each group are not provided. Given that the model wasn't trained on triple-residue mutations, and a purported strength of this method its ability to cleanly generalize to high-order mutations, this seems like an important omission.

**Questions:**

-     "We compute z(μ) = ft(xp) + ht..."

If h^t depends solely on t, why separate f^t and h^t?

-     “Sequence features from the Evoformer and Structure Module are aggregated as input to the decoder.”

Which sequence features exactly? How are they aggregated?

-     “To evaluate generalization on unseen proteins, we train on cDNA proteins with low similarity to those in all evaluation benchmarks.”

How exactly?

-     “We train on the cDNA display proteolysis dataset [61], which leverages a high throughput selection assay to extract noisy ∆∆G values for 100 mini-proteins totaling over 100,000 single and double mutations.”

Would be good to see some stats in the main paper. Protein length, MSA depth, etc.

-     “Our model demonstrates exceptional performance in prioritizing stabilizing double mutations over destabilizing ones, achieving a significantly higher normalized discounted cumulative gain of 0.43 compared to 0.25, as well as a superior detection precision of 0.16 compared to 0.10. Our model additionally improves classification metrics MCC and AUC by 0.02 and 0.03 respectively.”

If I understand this correctly, the baseline referenced here is just a version of your model where the heads at the end are replaced with simple addition? Why is that a meaningful comparison?

-     “While other methods also handle multiple mutations, they adopt a train and test split where unique mutations in the test set are not included in the training process. This inadvertently leads to training and testing on the same set of proteins [12, 28, 55]. PTMul proteins have at most 35% homology to the proteins used in these methods’ training. To fairly evaluate generalization to new proteins, we exclude these inflated comparisons from our study.”

Is this widely known? If not, this’ll require more specific justification, either in the main paper or the supplement.

- You finetune AlphaFold2 for the task. I'm kind of curious how well the method works if you simply train the MLPs with the embedding model frozen.

- Could you elaborate more on how AlphaFold2 was finetuned? There are some nontrivial implementation details here; e.g. how cropping is handled.

**Limitations:**

Authors include a brief discussion of limitations. They do not address potential negative societal impacts of their work (not that I think that would be warranted in this case).

---

> ### Author Rebuttal · Authors · 2023-08-09
>
> > While the model is evaluated on a mixture of double- and triple-residue mutations, individual results for each group are not provided… this seems like an important omission.
>
> Agreed. While our model performs similarly against the additive baseline on double mutations, it excels at higher-order mutations. We will include this analysis in the final version.
>
> Below are results for the double mutations in PTMul
> |                   | Spearman    | AUROC       | MCC         | RMSE        |
> |-------------------|-------------|-------------|-------------|-------------|
> | DDGun             | 0.28        | 0.63        | 0.22        | 2.23        |
> | DDGun3D           | 0.29        | 0.61        | 0.17        | 2.25        |
> | Additive Baseline | 0.52 (0.01) | 0.76 (0.01) | 0.30 (0.01) | 1.95 (0.01) |
> | MEM (ours)        | 0.50 (0.01) | 0.75 (0.01) | 0.34 (0.01) | 2.08 (0.01) |
>
> and triple and more mutations in PTMul
>
> |                   | Spearman    | AUROC       | MCC         | RMSE        |
> |-------------------|-------------|-------------|-------------|-------------|
> | DDGun             | 0.15        | 0.57        | 0.12        | 2.19        |
> | DDGun3D           | 0.19        | 0.60        | 0.17        | 2.20        |
> | Additive Baseline | 0.49 (0.04) | 0.79 (0.04) | 0.45 (0.06) | 2.14 (0.08) |
> | MEM (ours)        | 0.60 (0.01) | 0.86 (0.01) | 0.58 (0.02) | 1.99 (0.02) |
>
> > If h^t depends solely on t, why separate f^t and h^t?
>
> We tried both and found that adding it improves performance. We believe that separating them benefits training, as f^t is a deep network and h^t is an embedding that encodes each amino acid type separately.
>
> > “Sequence features from the Evoformer and Structure Module are aggregated as input to the decoder.” Which sequence features exactly? How are they aggregated?
>
> The Evoformer and Structure Module both output LxD representations for the input sequence where L is the sequence length and D is the hidden dimension. This is “Single repr. (r,c)” for Evoformer in AlphaFold2 Figure 1. More precisely, in the OpenFold code, it is `s` at the following lines: https://github.com/aqlaboratory/openfold/blob/1d878a1203e6d662a209a95f71b90083d5fc079c/openfold/model/evoformer.py#L823 and https://github.com/aqlaboratory/openfold/blob/1d878a1203e6d662a209a95f71b90083d5fc079c/openfold/model/structure_module.py#L753C1-L754C1
>
> The representations are normalized (LayerNorm) and added together before fed into the decoder. We experimented with the MSA and pair representations but decided not to use them to standardize the inputs from all backbones.
>
> > “To evaluate generalization on unseen proteins, we train on cDNA proteins with low similarity to those in all evaluation benchmarks.” How exactly?
>
> We compute the sequence similarity between the cDNA proteins and proteins in the validation set. We filter out any cDNA protein from our training set with higher than 30% sequence similarity with any test protein. This ensures that the proteins in the test set are unseen.
>
> > cDNA protein statistics.
>
> Great idea. The cDNA proteins average 56.1 amino acids in length with a maximum length of 72 and minimum length of 30 amino acids. The mean MSA depth is 7797 with a standard deviation of 6282. The maximum depth is 23525 and the minimum depth is 5. A comprehensive analysis of the dataset, experimental assay, filtering criteria are found in their paper [1].
>
> > The baseline referenced here is just a version of your model where the heads at the end are replaced with simple addition. Why is that a meaningful comparison?
>
> The impact of two mutations together on ΔΔG can differ from the impact of the two mutations performed separately (epistasis). Our finding that the model outperforms this additive baseline suggests that our model learns the interactions between single mutations.
>
> > Is it widely known that training and testing should have low homology?
>
> Yes, proteins with 35% or more sequence similarity are typically highly similar (e.g. same function in related organisms). Prior works have shown that training on proteins with substantial overlap with the test set leads to heavily overestimated performance [2].
> Many prior works build training (Q1744 [3]) and validation splits (s669 [4], t2837 [5]) with low similarity overlap. In this work, we filter our training set to keep low sequence similarity with all our validation sets.
>
> > Frozen embedding model
>
> We find that fine-tuning the AlphaFold2 backbone improves performance on S669.
> |           | Spearman | AUC  | MCC  | RMSE |
> |-----------|----------|------|------|------|
> | Freeze    | 0.48     | 0.72 | 0.20 | 1.46 |
> | Fine-tune | 0.56     | 0.76 | 0.37 | 1.36 |
>
> > Could you elaborate more on how AlphaFold2 was finetuned?
>
> We do not subsample the protein sequence during fine-tuning, as the proteins are shorter than the crop size. The MSA sampling is performed randomly.
>
> Many implementation details are hard to explain in plain English. We are happy to publish the code upon acceptance.
>
> [1] Tsuboyama K., Dauparas, J., Chen, J., Mangan N., Ovchinnikov S., Rocklin, G.. Mega-scale experimental analysis of protein folding stability in biology and protein design.
>
> [2] Montanucci L, Savojardo C, Martelli PL, Casadio R, Fariselli P. On the biases in predictions of protein stability changes upon variations: the INPS test case.
>
> [3] Li, B., Yang, Y.T., Capra, J.A., Gerstein, M.B.: Predicting changes in protein thermodynamic stability upon point mutation with deep 3d convolutional neural networks.
>
> [4] Pancotti, C., Benevenuta, S., Birolo, G., Alberini, V., Repetto, V., Sanavia, T., Capriotti, T. and Fariselli, P. Predicting protein stability changes upon single-point mutation: a thorough comparison of the available tools on a new dataset.
>
> [5] Diaz, D.J., Gong, C., Ouyang-Zhang, J., Loy, J.M., Wells, J.T., Yang, D., Ellington, A.J., Dimakis, A., Klivans, A.R.: Stability oracle: A structure-based graph-transformer for identifying stabilizing mutations.

---

> > ### Comment · Reviewer_AB6d · 2023-08-10
> >
> > >Yes, proteins with 35% or more sequence similarity are typically highly similar (e.g. same function in related organisms). Prior works have shown that training on proteins with substantial overlap with the test set leads to heavily overestimated performance [2]. Many prior works build training (Q1744 [3]) and validation splits (s669 [4], t2837 [5]) with low similarity overlap. In this work, we filter our training set to keep low sequence similarity with all our validation sets.
> >
> > I understand that training and validation proteins should have low sequence similarity; my question was whether it's widely known that the papers you mention ([12, 28, 55]) have sketchy validation sets. Unless that claim is documented somewhere, you'll need to provide more details to prove it before you can toss out their reported metrics.
> >
> > The rest looks good. If the above is resolved, I'll raise my score from 6 to 7.

---

> > > ### Author Response · Authors · 2023-08-11
> > >
> > > > I understand that training and validation proteins should have low sequence similarity; my question was whether it's widely known that the papers you mention ([12, 28, 55]) have sketchy validation sets. Unless that claim is documented somewhere, you'll need to provide more details to prove it before you can toss out their reported metrics.
> > >
> > > Thank you for clarifying the question. We will provide these details in the final version.
> > >
> > >  “Predicting the effect of single and multiple mutations on protein structural stability”
> > >  [12] generates splits at the residue level, so similarity between training and validation proteins is not accounted for. In Section 4.1.4., “data was split into these sets under the constraint that each unique wild type mutation combination appeared only in a single set.”
> > >
> > > Maestro [28] performs cross validation (see Table 1 in [28]). We could not find the split curation in the paper, but found the same protein in multiple folds in their data release. https://bmcbioinformatics.biomedcentral.com/articles/10.1186/s12859-015-0548-6#additional-information
> > >
> > > Dynamut2 [55] generates splits at the mutation level. In Section 4.1 paragraph 2, “Our final dataset comprised 1,098 entries (710 destabilizing and 388 stabilizing) (Figure S4), which were randomly split into train and test sets comprising 872 and 227 entries, respectively.” Their train and test data for multiple mutations contain overlapping pdb IDs (e.g. 1ACB) at
> > > https://biosig.lab.uq.edu.au/dynamut2/data
> > >
> > > We categorized the Dynamut2 test set based on whether the sample’s PDB ID appears in the training set (found at https://biosig.lab.uq.edu.au/dynamut2/data). The counts and errors are detailed below. Errors are noticeably higher for the PDB IDs that have not been seen during training compared to those that have been seen.
> > > This suggests that their test set performance might not be representative of their model’s performance on new proteins.
> > > Please note that in this context, we are not using a strict sequence similarity filter, but rather a more lenient matching filter.
> > > |            | Num. PDB | RMSE |
> > > |------------|----------|------|
> > > | Full Test  | 226      | 1.66 |
> > > | Seen PDB   | 213      | 1.59 |
> > > | Unseen PDB | 13       | 2.52 |
> > >
> > > [12] Dehghanpoor, R., Ricks, E., Hursh, K., Gunderson, S., Farhoodi, R., Haspel, N., Hutchinson, B., Jagodzin- ski, F.: Predicting the effect of single and multiple mutations on protein structural stability.
> > >
> > > [28] Laimer, J., Hofer, H., Fritz, M., Wegenkittl, S., Lackner, P.: Maestro-multi agent stability prediction upon point mutations.
> > >
> > > [55] Rodrigues, C.H., Pires, D.E., Ascher, D.B.: Dynamut2: Assessing changes in stability and flexibility upon single and multiple point missense mutations.

---

> > > > ### Comment · Reviewer_AB6d · 2023-08-11
> > > >
> > > > LGTM. Score raised.

---

### Official Review · Reviewer_KhLn · 2023-07-06

**Soundness:** 3 good
**Presentation:** 3 good
**Contribution:** 2 fair
**Rating:** 3
**Confidence:** 4

**Summary:**

This work predicts changes in thermodynamic stability for single or higher-order mutations on top of AlphaFold2 modules. The proposed model leverages linear aggregation of mutational scores on all possible sites in the latent space to decode $\Delta\Delta G$ value for deep mutations in parallel.


**Strengths:**

- The workflow is easy to understand, which proposes a lightweight solution for an important direction of research.
- Compared to most existing work, the proposed method 'runs a strong backbone only once and a lightweight decoder N times in parallel'.

**Weaknesses:**

- The innovation is incremental. The main algorithm relies heavily on the existing AlphaFold2 model.
- While the authors claim the main contribution is that they designed 'a simple, parallel decoding algorithm', this is not the first method that tries to decode all possible mutational scores from a large latent representation (a $L\times 20$ matrix in this paper). See, for instance, https://arxiv.org/pdf/2304.08299.pdf.
- Notations are poorly explained. For instance, the meaning of p, t, and superscription A, Y are undefined on page 4.
- References are missed in many places (for instance, ESM2 in Table 3). Also please define abbreviations properly when they first appear (even when they are used widely). For example, AUC and MCC are on page 6.
- A conclusion section is missed at the end.

**Questions:**

- How is the ratio '3%' observed in line 176?
- Similarly, why $\Delta\Delta G<-0.5$ are considered stabilizing? Any reference here?
- Since the authors use ProteinGym, is the proposed method applicable to amino acid addition and deletion?

**Limitations:**

Although the authors discussed limitations of the research in Section 6, they focused mainly on the technical limitations. No negative society impact was mentioned.

---

> ### Author Rebuttal · Authors · 2023-08-09
>
> > “The innovation is incremental. The main algorithm relies heavily on the existing AlphaFold2 model.”
>
> We experimented with multiple backbones, including AlphaFold2, ESM2, MSA-Transformer (see Table 6a). Among these backbones, AlphaFold2 happens to be the strongest performing backbone. We present a new model paradigm that predicts the effects of all mutations, rather than a single mutation or position as in prior works [1,2,3,4]. We see our contribution as orthogonal to the choice of backbone.
>
> Earlier works found that AlphaFold2 is not sensitive to single mutations [5], concluding “AlphaFold may not be immediately applied to other problems or applications in protein folding.” To the best of our knowledge, we are the first to apply AlphaFold2 for predicting ΔΔG values of mutations.
>
> > This is not the first method that tries to decode all possible mutational scores from a large latent representation (a  L × 20  matrix in this paper). See, for instance, https://arxiv.org/pdf/2304.08299.pdf.
>
> Thanks for sharing “Accurate and Definite Mutational Effect Prediction with Lightweight Equivariant Graph Neural Networks'' by Bingxin Zhou, Outongyi Lv, Kai Yi, Xinye Xiong, Pan Tan, Liang Hong and Yu Guang Wang. Per NeurIPS guidelines, this paper is considered contemporaneous work as it was arxiv’ed 34 days before the NeurIPS paper deadline. We are happy to discuss it as such in the final version.
>
> > Notations are poorly explained. For instance, the meaning of p, t, and superscription A, Y are undefined on page 4.
>
> We decided to explain all the notations in the preliminary (p,t,A,Y are all defined in lines 113-115). We can see that this might be confusing and will work on clearing this up.
>
> > References are missed in many places (for instance, ESM2 in Table 3). Also please define abbreviations properly when they first appear (even when they are used widely). For example, AUC and MCC are on page 6.
>
> Thank you, we will add them.
>
> > A conclusion section is missed at the end.
>
> We omitted a conclusion due to space constraints. We will include one in the additional page allotted for the final version.
>
> > How is the ratio '3%' observed in line 176?
>
> Line 176 states "Only 3% of the mutation sets in our training set are stabilizing.” This is a dataset statistic. Our training set consists of mutations and their corresponding ΔΔG value. The percentage of mutations with ΔΔG < -0.5 kcal/mol is 3%.
>
> > Why ΔΔG < − 0.5 are considered stabilizing? Any reference here?
>
> We follow Benevenuta et al. [6] in categorizing mutations with a 0.5 kcal/mol decrease in free energy as stabilizing. We will add the reference to the final version.
>
> > Since the authors use ProteinGym, is the proposed method applicable to amino acid addition and deletion?
>
> We have not considered insertions and deletions yet. Our primary application of protein engineering modifies an existing protein only slightly to increase thermodynamic stability. Insertions and deletions cause a shift in the entire sequence, leading to global changes in the protein structure.
>
> [1] Benevenuta, S., Pancotti, C., Fariselli, P., Birolo, G., Sanavia, T.: An antisymmetric neural network to predict free energy changes in protein variants.
>
> [2] Li, B., Yang, Y.T., Capra, J.A., Gerstein, M.B.: Predicting changes in protein thermodynamic stability upon point mutation with deep 3d convolutional neural networks.
>
> [3] Umerenkov, D., Shashkova, T.I., Strashnov, P.V., Nikolaev, F., Sindeeva, M., Ivanisenko, N.V., Kardymon, O.L.: Prostata: Protein stability assessment using transformers.
>
> [4] Diaz, D.J., Gong, C., Ouyang-Zhang, J., Loy, J.M., Wells, J.T., Yang, D., Ellington, A.J., Dimakis, A., Klivans, A.R.: Stability oracle: A structure-based graph-transformer for identifying stabilizing mutations.
>
> [5] Pak, M.A., Markhieva, K.A., Novikova, M.S., Petrov, D.S., Vorobyev, I.S., Maksimova, E.S., Kondrashov, F.A., Ivankov, D.N.: Using alphafold to predict the impact of single mutations on protein stability and function.
>
> [6] Benevenuta S., Birolo G., Sanavia T., Capriotti E., Fariselli P. Challenges in predicting stabilizing variations: An exploration.

---

> > ### Author Response · Authors · 2023-08-17
> >
> > Is there anything else the reviewer would need to know to raise the final rating?

---

### Decision · Program_Chairs · 2023-09-21

**Decision:**

Accept (poster)

**Comment:**

This paper proposed an efficient method to perform one and double mutations of proteins in a single pass on top of the AlphaFold module. Experimental results on multiple data sets show the effectiveness of the proposed approach.

After the discussion, all the reviewers like the simplicity of the proposed approach and agree this is a good contribution in solving a practical and important problem.